# N-3 PUFA and Pregnancy Preserve C-Peptide in Women with Type 1 Diabetes Mellitus

**DOI:** 10.3390/pharmaceutics13122082

**Published:** 2021-12-04

**Authors:** Josip Delmis, Marina Ivanisevic, Marina Horvaticek

**Affiliations:** 1Clinical Department of Obstetrics and Gynecology, University Hospital Centre Zagreb, School of Medicine, University of Zagreb, 10000 Zagreb, Croatia; marina.ivanisevic@pronatal.hr; 2Institute Rudjer Boskovic, 10000 Zagreb, Croatia; marina.horvaticek@gmail.com

**Keywords:** pregnancy, type 1 diabetes mellitus, beta-cell, insulin, C-peptide, hypoglycemia, n-3 PUFA, placenta, fetus

## Abstract

Type 1 diabetes (T1DM) is an autoimmune disease characterized by the gradual loss of β-cell function and insulin secretion. In pregnant women with T1DM, endogenous insulin production is absent or minimal, and exogenous insulin is required to control glycemia and prevent ketoacidosis. During pregnancy, there is a partial decrease in the activity of the immune system, and there is a suppression of autoimmune diseases. These changes in pregnant women with T1DM are reflected by Langerhans islet enlargement and improved function compared to pre-pregnancy conditions. N-3 polyunsaturated fatty acids (n-3 PUFA) have a protective effect, affect β-cell preservation, and increase endogenous insulin production. Increased endogenous insulin production results in reduced daily insulin doses, better metabolic control, and adverse effects of insulin therapy, primarily hypoglycemia. Hypoglycemia affects most pregnant women with T1DM and is several times more common than that outside of pregnancy. Strict glycemic control improves the outcome of pregnancy but increases the risk of hypoglycemia and causes maternal complications, including coma and convulsions. The suppression of the immune system during pregnancy increases the concentration of C-peptide in women with T1DM, and n-3 PUFA supplements serve as the additional support for a rise in C-peptide levels through its anti-inflammatory action.

## 1. Introduction

Type 1 diabetes is a condition caused by autoimmune damage to the insulin-producing β-cells of the pancreatic islets, usually leading to severe endogenous insulin deficiency [1]. The natural course of type 1 diabetes includes four very different stages of the disease:preclinical autoimmunity towards beta cells with progressive decline in insulin production;the onset of clinical diabetes;transient remission; andclinically pronounced diabetes with acute and chronic complications (Figure 1).

Early introduction of insulin into the treatment of type 1 diabetes, in addition to the primary effect of good metabolic control, appears to have an additional effect on delaying the destruction of β-cells. There are considerations according to which the use of exogenous insulin in the early phase of the disease puts β-cells in a state of relative metabolic rest and a lower load on insulin production, which makes them resistant to destruction [2]. Intensive insulin therapy effectively delays the onset and slows the progression of diabetic retinopathy, nephropathy, and neuropathy in patients with T1DM [3].

The goal of treating women with type 1 diabetes who plan a pregnancy is to achieve optimal glycemic control before and during pregnancy in order to reduce the incidence of miscarriage, congenital malformations, macrosomia, and neonatal complications. Poor metabolic control in pregnant women with type 1 diabetes mellitus is associated with an increased risk of miscarriage, preeclampsia, congenital malformations, asphyxia, macrosomia, and perinatal morbidity and mortality [4,5,6]. A successful perinatal outcome requires intensive clinical care in order to achieve normoglycemia before conception and during pregnancy. Proper diet and intensive insulin therapy are effective in achieving the desired glycemic targets. Good metabolic control is achieved when the venous plasma glucose values are between 4.0 and 5.0 mmol/L before a meal and 5.0 and 7.8 mmol/L after a meal and HbA1c values are lower than 6.0% (<42 mmol/mol) [7]. Significantly improved metabolic control of pregnant women with T1DM, intensive antenatal and neonatal care, more frequent antenatal ultrasound examinations, and cardiotocographic monitoring with a more liberal attitude towards cesarean delivery are mirrored by perinatal mortality rate below ten per mill; it shows a steady downward trend [5,6,7].

Intensive glycemic control improves pregnancy outcomes; however, it increases the risk of hypoglycemia [8,9] and causes maternal complications, including coma, convulsions, and death [10]. Hypoglycemia affects many pregnant women with T1DM and is several times more common than that outside of pregnancy [11,12].

## 2. Materials and Methods

The literature was reviewed using relevant databases (Pubmed/Medline, Scopus, Science Direct). The following inclusion criteria were considered: studies that investigated hypoglycemia (hypoglycemia in pregnant women with type 1 diabetes), n-6 PUFA and n-3 PUFA (in pregnant women; in pregnant women with type 1 diabetes), supplementation of EPA and DHA (in pregnant women with type 1 diabetes) C-peptide (in pregnant women with type 1 diabetes; in nonpregnant people), the immune system in pregnancy, vitamin D T1DM and relevant observational, randomized control trial and review articles were selected to provide information and data for the text (published from 2000 to 2021). Only papers written in English were selected.

### Exclusion Criteria

We excluded papers with gestational and type 2 diabetes. Articles on immunosuppressive drugs (corticosteroids, azathioprine, methotrexate, cyclophosphamide, and cyclosporine) associated with C-peptide preservation were also excluded.

## 3. Hypoglycemia in Pregnant Women with T1DM

Known risk factors for hypoglycemia in pregnancy are the duration of type 1 diabetes mellitus, history of previous severe hypoglycemia, hypoglycemia unawareness, changes in insulin treatment, and HbA1c < 6.5% (48 mmol/mol). Reducing the risk of hypoglycemia is a major challenge for physicians taking care of pregnant women with T1DM.

The International Hypoglycaemic Research Group (IHSG) unequivocally considers a glucose concentration of <3.0 mmol/L as a hypoglycemic value detected by plasma self-monitoring, continuous glucose monitoring (at least 20 min), or laboratory plasma glucose measurement [13]. IHSG considers a glucose concentration of <3.0 mmol/L low enough to indicate severe, clinically significant hypoglycemia [13]. The same group suggested that a glucose value of 3.9 mmol/L or less should indicate possible hypoglycemia. Severe hypoglycemia is considered a hypoglycemic episode that requires assistance from another person to treat.

### 3.1. Insulin and C-Peptide

Insulin is a low-molecular-weight protein (5808 daltons) secreted as a primary response to elevated blood glucose levels [14]. Insulin is the primary hormone that controls glucose metabolism and facilitates the entry of glucose into cells [14]. Some nerve stimuli and elevated concentrations of other energy sources, such as amino acids and fatty acids, stimulate insulin secretion [15]. The central role of insulin is to control the entry of glucose from the blood into the cells and its utilization in peripheral tissues. Insulin also participates in lowering blood glucose by stimulating glycolysis, inhibiting gluconeogenesis and glycogenolysis in the liver. Insulin’s action is counter-regulated by hormones that increase glucose concentration, such as glucagon, adrenaline, growth hormone, and cortisol. Healthy pancreatic beta cells produce and secrete as much insulin as is needed to control glucose homeostasis [15].

Insulin is synthesized from a pre-proinsulin precursor molecule. C-peptide is a peptide consisting of 31 amino acids that, in the proinsulin molecule, connects the amino end of the alpha chain and the carboxyl end of the beta chain of insulin, hence the name (connecting peptide (C-peptide)) [16]. Insulin and C-peptide are excreted together in equimolar amounts into the portal circulation. About 50% of insulin is metabolized (absorbed) in the liver during the first pass; this is called the “first-pass effect.” Residual insulin plays an important regulatory role in peripheral tissues. Unlike insulin, C-peptide is not absorbed by the liver and has a longer half-life (about 30 min) than insulin (about 4 min) and is suitable for measuring the concentration of endogenous insulin in the peripheral circulation [16]. The close association of C-peptide in the systemic circulation and endogenous insulin in the portal system has been well established [16,17]. C-peptide concentration under standardized conditions in the peripheral circulation is the most appropriate measure of endogenous insulin secretion and β-cell function in pregnant women with T1DM.

From the time when a clinical diagnosis of T1DM is made, patients typically retain a limited ability to produce endogenous insulin for several months or years [18,19,20] (Figure 1). Excretion is gradually lost as β-cell function is selectively destroyed. Autoimmune destruction of β-cells is more pronounced/aggressive when T1DM develops in childhood and youth, while this process is longer, more variable, and beta cells more slowly deteriorate entirely in patients with T1DM later in life [21,22,23].

### 3.2. The Immune System in Pregnancy

Pregnancy is a unique event in which the genetically and immunologically different fetus survives until full term without being rejected from its mother’s immune system [24]. It is associated with partial suppression of the immune and pro-inflammatory system, which is why autoimmune diseases such as diabetes often go into remission during pregnancy [25].

The maternal immune system undergoes significant changes, including developing specific pathways to protect the fetus from the mother’s cytotoxic attack. One mechanism decreases the expression of classical HLA class I molecules, while other mechanisms are associated with an altered balance of Th1 and Th2 [25]. During pregnancy, cellular immune function and pro-inflammatory Th1 cytokines (e.g., IL-2, TNF-α, and INF-γ) are suppressed, while humoral immunity and the production of anti-inflammatory Th2 cytokines (e.g., IL-4 and IL-10) are enhanced [25]. Such a pattern of immune function is reversed in the postpartum period [25]. A partial reduction in the activity of the inflammatory immune system leads to an improvement in various autoimmune diseases during pregnancy, including diabetes. In type 1 diabetes, these changes are expressed by the growth and functional modification of the Langerhans pancreatic islets. The most significant change that these islets undergo during pregnancy is the increased secretion of insulin. Numerous animal models have shown that β-cell mass increases three to four times during pregnancy with significant hypertrophy and β-cell proliferation [26,27,28]. In a pilot study, Amouyal C et al. found that higher levels of regulatory T cells and IL-2 improved endogenous insulin secretion in pregnant women with T1DM [29]. The authors [30] showed that C-peptide concentrations increase gradually during pregnancy in women with type 1 diabetes.

Meeck CL et al. [31] have measured maternal serum C-peptide concentrations at 12, 24 and 34 weeks of gestation in 127 pregnant women with type 1 diabetes and cord blood C-peptide concentrations. In 74 (58%) pregnant women, C-peptide was not detected; in 22 (17%), it was confirmed at the beginning of pregnancy, and in 31 (24%), it was detected in the 34th week of pregnancy. Neonates born to the mothers in whom C-peptide was detected at 34 weeks of gestation had elevated cord blood C-peptide and more frequent neonatal complications than others. Based on the results, the authors suggest a transfer of C-peptide from fetal to maternal serum without the regeneration of pregnancy-related beta cells. Pregnant women with detected C-peptide had better-regulated glycemia, fewer hypoglycemia events according to CGM, reduced total insulin dose, and lower incidence of macrosomia [31]. According to other authors’ research, it is plausible that pregnancy yields immunological tolerance and stimulates endogenous insulin production in women with type 1 diabetes mellitus [30]. Due to the small number of participants, the authors [32] did not prove an increase in the concentration of C-peptide during pregnancy, which, nonetheless, does not contradict the findings of other authors.

## 4. Fatty Acids

Fatty acids (FA) are found in most lipids and have several physiologically significant roles in mammalian tissues, such as structural roles, storage, and energy production, and as precursors of biologically active substances. Fatty acids are constituent units of phospholipids and glycolipids and are, therefore, a fundamental component of cell membranes that give the cell its structural integrity and delimit the organelles [32]. They are rarely found in free form in the body due to their detergent and cytotoxic effects but are generally esterified into larger molecules such as phospholipids (PL) and triacylglycerols (TAG). They are divided based on the degree of saturation:saturated fatty acids (SFA), which do not contain double (unsaturated) bonds;monounsaturated fatty acids (MUFAs) in which there is one double bond between carbon atoms; andpolyunsaturated fatty acids (PUFA) in which there are several double bonds.

The human body can produce saturated and monounsaturated fatty acids with a double bond on the ninth carbon atom counting from the end of the chain (n-9 unsaturated fatty acids). However, due to the lack of the Δ-12 and Δ-15 desaturase enzymes, humans cannot de-novo synthesize the n-6 and n-3 fatty acid families [32]. Linoleic acid (LA, C18: 2n-6), from which an n-6 series of unsaturated fatty acids are further synthesized in the body, and α-linolenic acid (ALA, C18: 3n-3), from which n-3 series of unsaturated fatty acids are synthesized acids, must, therefore, be taken into the body through food and are called essential fatty acids (EFA). The biosynthesis of long-chain polyunsaturated fatty acids (LCPUFA) from precursors involves a series of alternating activities of the enzymes elongase and desaturase, with competition between n-6 and n-3 acids for desaturation enzymes [33]. However, the effectiveness of endogenous conversion of ALA from food to EPA, DPA, and DHA in the adult human body is not significant [33]. Due to the low n-3 PUFA intake, supplementation of these fatty acids is recommended to gain and preserve cardiovascular health and establish the normal neurological development of fetuses and newborns [34].

### Inflammatory Processes and n-3 Polyunsaturated Fatty Acids

Long-chain PUFAs such as AA, EPA, and DHA in the body can originate directly from the food consumed or are synthesized endogenously from their food-derived precursors [35]. The main representative of n-3 PUFA is ALA, found in green leafy vegetables and some seeds, nuts, and legumes. It can be metabolized in a limited capacity to EPA and DHA, two n-3 PUFAs whose primary source is algae, fish, and seafood [35]. The most abundant n-6 PUFA in the diet is LA, which is primarily found in nuts, seeds, and vegetable oils. It is a precursor to AA synthesis. Since ALA and LA compete for crucial enzymes involved in fatty acid metabolism and conversion to pro-inflammatory or anti-inflammatory eicosanoids, it is important to investigate the effect of co-intake of n-3 and n-6 PUFAs [36]. Prostanoids are cyclooxygenase-formed eicosanoids and include prostaglandins (PG), prostacyclins (PGI), and thromboxanes (TX). Leukotrienes (LT) and lipoxins (LX) are eicosanoids formed by lipoxygenase. Precursors, especially AA and EPA, can be derived directly from food or are metabolically derived from LA and ALA. Fatty acids for eicosanoid synthesis are usually released from the sn-2 position of phospholipids from the cell membrane (the site where PUFAs are most commonly esterified) by the action of the enzyme phospholipase A2 (PLA2) [36]. Therefore, the activity of PLA2 is a limiting factor in the rate of eicosanoid formation from AA and EPA. Since PLA2 has the same affinity for AA and EPA, the fatty acid composition of phospholipids depends on the ratio of released n-3 and n-6 PUFAs used as starting compounds for eicosanoid synthesis [36]. With the Western diet, significant amounts of AA are introduced into the body, which is consequently the most abundant PUFA in membrane phospholipids, so the most common forms of PG series 2 in the human body are eicosanoids synthesized from fatty acids n-6 (AA) and n-3 (EPA) order they interact antagonistically. AA metabolism produces pro-inflammatory eicosanoids, prostaglandins, and thromboxanes of series 2 (PG2 and TX2) and leukotrienes of series 4 (LT4).

In contrast, EPA metabolism produces anti-inflammatory eicosanoids, prostaglandins, and thromboxanes of series 3 (PG3 and TX3) and leukotrienes of series 5 (LT5) [36]. Therefore, an elevated AA/EPA or AA/DHA ratio in the body may indicate inflammatory processes. Eicosanoids are considered to be a link between n-3 PUFA, inflammation, and immunity [37]. In addition to anti-inflammatory effects by suppressing LT4 synthesis, n-3 PUFAs suppress the excessive ability of monocytes to synthesize interleukin-1 (IL-1) and tumor necrosis factor (TNF). The anti-inflammatory properties of n-3 PUFA, especially EPA and DHA, result from the competition with AA as a substrate for cyclooxygenase (COX) and 5-lipoxygenase. Hence, EPA and DHA increase the synthesis of LT5 (which has a weaker inflammatory effect than LT4) and decrease the synthesis of LT4 derived from AA [35]. PG3 is a potent vasodilator and inhibitor of platelet aggregation like PG2, while TX3 is a significantly weaker platelet aggregator with minimal vasoconstrictive effect, unlike TX2. These changes in PG, TX, and LT production alter the balance of their activities, such as vasodilation and inhibition of platelet aggregation, resulting in decreased inflammatory activity. DHA is a precursor of resolvins, protectins, and maresins, characterized by anti-inflammatory effects [36]. Several recent studies show that changes in the amount and type of fatty acids in the diet can affect the formation of various eicosanoids.

Replacement of food containing oils with predominant n-6 fatty acids with oils with prevailing n-3 fatty acids results in increased synthesis of less potent eicosanoids than those derived from n-6 fatty acids.

## 5. n-3 PUFA and C-Peptide Preservation

### 5.1. n-3 PUFA and C-Peptide Preservation in Pregnant Women with T1DM

A prospective randomized placebo-controlled clinical trial [37] included 90 pregnant women with T1DM. Forty-seven of them took n-3 supplementation (60 mg of EPA and 308 mg of DHA) from the 9th week of pregnancy, while 43 took a placebo and were on a regular diabetic diet. In this clinical trial, the duration of T1DM was between 5 and 30 years, and pregnant women were divided into two subgroups. The first subgroup included 36 pregnant women with a duration of T1DM from 5 to 10 years, where 21 were on n-3 supplementation, and nine were on placebo. The second subgroup included 54 pregnant women with a duration of T1DM from 11 to 30 years, while 26 were on n-3 supplementation and 28 were on placebo. In both fasting C-peptide tended to increase during pregnancy. The increase in FC-peptide during pregnancy was statistically significant in the intervention group, while in the control group, there was no statistically significant increase in FC-peptide in the third trimester compared to the FC-peptides values from the first and second trimester [37] (Table 1).

In both subgroups of pregnant women, based on the duration of T1DM (5–10 and 11–30 years), a statistically significant difference was found in the intervention group between FC-peptide values in the first and third trimester, while this difference was not found in the control group. A significant reduction in the dose of long-acting insulin was found in both study subgroups [37].

The results unequivocally demonstrate the importance of supplementation with EPA and DHA in pregnant women with T1DM. Measurement of C-peptide concentration provides a validated way to quantify endogenous insulin secretion. As the C-peptide does not cross the placenta in either direction, the C-peptide values detected during pregnancy are derived from maternal beta cells and not from the fetus. The increase in C-peptide concentration in both groups of pregnant women is due to suppression of the pro-inflammatory system during pregnancy. Therefore, the maternal chances of having a genetically and immunologically diverse fetus are significantly increased, and EPA and DHA have an additional synergistic impact as they equally produce anti-inflammatory effects. After childbirth, there is a rapid decrease in C-peptide concentration, which indicates the active role of the placenta in increasing the concentration of C-peptide during pregnancy [30].

### 5.2. n-3 PUFA and C-Peptide Preservation in Nonpregnant Patient with T1DM

Mayer-Davis EJ. et al. included surveillance of a large number (1316) of young people with eating habits in a prospective two-year study called SEARCH for Diabetes in Youth. [38]. Participants were measured for DHA and EPA concentrations in venous plasma and venous serum FC-peptide. An association was found between the concentration of n-3 PUFA and FC-peptide. Higher concentrations of EPA and DHA were associated with higher concentrations of FC-peptide. According to FFQ (Food Frequency Questionnaires) data, an association was found between increased leucine intake and higher FC-peptide concentrations [38].

The result of this study strongly confirms the effect of n-3 PUFA on C-peptide conservation in young individuals with T1DM (Table 1).

## 6. Vitamin D Has a Protective Effect in Preventing T1DM

Studies show that vitamin D can have a protective effect in preventing the formation of T1DM [39,40].

The active form of vitamin D (D3, 1.25 dihydrocolecalciferol) has an immunomodulatory role in preventing type 1 diabetes through the activation of β-cells. At the level of pancreatic islets, vitamin D3 reduces proinflammatory cytokines (IL-1, IL6), making β-cells less chemoattractive and more resistant to inflammatory mediators, which ultimately results in a decrease in total T-lymphocytes and an increase in the number of regulatory (helper/suppressor) mononuclear cells, which prevent the autoimmune process. Therefore, dietary vitamin D intake during pregnancy and early childhood is considered to reduce the risk of developing autoimmune diabetes. Hyppönen E et al. [40] proved, in a long-term prospective study, that vitamin D supplementation, started in the first year of life in children, reduces the incidence (prevalence) of T1DM compared to those without it (RR 0.12, 95% CI 0.03–0.05) [40].

## 7. C-peptide, Insulin Doses, and Glycemic Control

The most reliable indicator of maintaining beta-cell function is the concentration of C-peptides in the blood [16]. Measurement of the C-peptide concentration provides a validated method for quantifying secreted endogenous insulin. The close association between C-peptide in the systemic circulation and endogenous insulin in the portal system is well established [16]. The C-peptide concentration gradually increases during pregnancy, independent of blood glucose concentration, in pregnant women suffering from type 1 diabetes mellitus [29,38]. A significant increase in fasting C-peptide occurred in the third trimester of pregnancy compared to the first trimester [37].

Preservation of β-cell function is associated with a lower risk of hypoglycemic events, as well as lower HbA1c and less frequent microvascular complications [17]. Nutritional factors previously identified as potentially protective against the development of type 1 diabetes [38], or those associated with insulin production may also be valuable for the preservation of the β-cell function.

Ilić et al. [41] found an association between higher C-peptide concentrations and lower total insulin doses in the first trimester of pregnancy. A lower prevalence of severe hypoglycemia and a decline in total insulin units during pregnancy were associated with increased endogenous insulin secretion [41]. Pregnancy induces an increase in fasting C-peptide concentration in most women with diabetes mellitus but is more pronounced in those with a shorter duration of diabetes [38]. A lower prevalence of severe hypoglycemia and a decline in required total insulin units during pregnancy were associated with increased endogenous insulin secretion. C-peptide serves as the mediator in the correlation between the duration of type 1 diabetes and the required total insulin dose [42,43] (Table 1).

Hypoglycemia in pregnant women with type 1 diabetes most often occurs due to insulin overdose or untimely insulin delivery, missed or reduced (insufficient) meals, emesis or hyperemesis in the first trimester of pregnancy, gastroparesis, or increased glucose consumption during and immediately after exercise [8]. Reducing the risk of hypoglycemia is a major challenge for doctors caring for pregnant women with T1DM. Technological advances in insulin production (e.g., analogues), insulin delivery devices (e.g., insulin pumps) and glucose monitoring devices have contributed to overall metabolic improvement, but in clinical practice, severe hypoglycemia rates, especially in pregnant women, remain high. Continuous glucose monitoring (CGM) is a method of continuous monitoring of glucose levels in interstitial fluid which was coined for improving metabolic control and expected result of glycemic monitoring is a reduction in hyperglycemia and a reduction in low glucose levels including symptomatic hypoglycemia [7,44].

## 8. Conclusions and Recommendation

Intake of n-3 long-chain polyunsaturated fatty acids affects the health of the mother, fetus, and child.

N-3 PUFA increases endogenous insulin production, resulting in reduced daily insulin doses, better metabolic control, reduced risk of long-term diabetic complications, and adverse effects of intensified insulin therapy, primarily hypoglycemia. There is evidence that n-3 PUFA prevents preterm birth before 34 gestational weeks [45].

In addition to the beneficial effects for the mother, n-3 PUFA affects the health of the child. Docosahexaenoic acid is incorporated into the fetal brain and retinal phospholipids, resulting in a high degree of fluidity and flexibility of neural and endothelial membranes. It is involved in neurotransmission, regulates ion channel activity and gene expression, and can be metabolized to neuroprotective metabolites [46]. Offspring from the mothers taking docosahexaenoic acid supplementation had higher IQs than those in the control group [47].

Due to the many beneficial effects for both mother and fetus, n-3 PUFA supplementation is recommended in women who plan pregnancy and pregnant women with T1DM. The European Food Safety Authority (EFSA) allows reaching the Dietary Reference Value for LC-PUFA (250 mg EPA plus DHA, plus an additional 100–200 mg of DHA) by food and supplements [48].

## Figures and Tables

**Figure 1 pharmaceutics-13-02082-f001:**
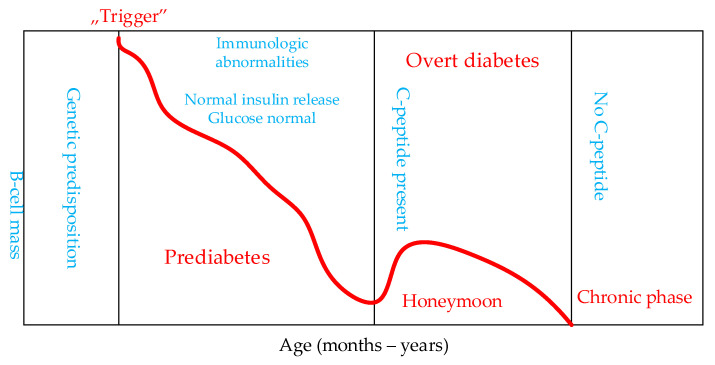
Stages in the development of type 1 diabetes mellitus. Phase 1 is the preclinical phase with the development of antibodies (ICA, IAA, GAD, ZnT8). Phase 2 is the preclinical phase which can last for months and years. Phase 3 is the onset of clinical diabetes with transient remission (“Honeymoon”). Phase 4 is clinically advanced diabetes with acute and chronic complications.

**Table 1 pharmaceutics-13-02082-t001:** Reviews of studies that evaluate C-peptide levels in pregnant women and young population with type 1 diabetes.

Reference	Study Population	No of Participants	Evaluated Parameters	Results/Conclusions
Ilic, S. [41]	Pregnant women T1DM At 10 w of gestation	10	Fasting C-peptide	C-peptide non-detectable before pregnancy to detectable at 10 weeks of gestation.
Nielsen L. [30]	Pregnant women T1DM Early and late pregnancy, postpartum	108; two groups based on serum glucose, >5.0 or <5.0 mmol/L	C-peptide, glucose, placental GH, IGF-I.	C-peptide in women with long-term T1DM C-peptide in early pregnancy: raised up to 97% by 33 weeks gestation.
Murphy, HR [32]	Pregnant women T1DM Early and late pregnancy	10	Fasting or meal- stimulated C-peptide concentration.	No change in fasting or meal-stimulated plasma C-peptide from early to late pregnancy.
Mayer-Davis, EJ. [38]	Young people (up to 20 years)/T1DM	1316	EPA and DHA, vitamins D E. Intake of leucine	Intake of n-3 PUFA sustained β-cell function, higher DHA and EPA were associated with higher fasting C-peptide.
Horvaticek, M. [37]	Pregnant women T1DM Three trimesters of pregnancy	90; two groups, n-3 supplementation or placebo	Fasting C-peptide, FPG EPA and DHA in maternal and cord blood serum.	n-3 PUFAs and pregnancy stimulates the production of endogenous insulin in women with T1DM Rise in FC-peptide during pregnancy was significant in exposed group
Amouyal, C. [29]	Pregnant women T1DM Three trimesters of pregnancy and post-partum	13	Clinical, immunological and diabetes parameters.	One group (*n* = 7) rise in C-peptide between the 2nd and 3rd trimesters, while second group had no β-cell function improvement during pregnancy.
Meek, L. [31]	Pregnant women T1DM Three trimesters of pregnancy	127; three groups based on detection of maternal C-peptide	Serum C-peptide concentration in a maternal and cord blood samples	Most women had undetectable C-peptide throughout pregnancy. Second group with detectable C-peptide characterized by lower BMI, later onset, shorter duration of T1DM improved glycemic control. A third group had the first appearance of C-peptide in maternal serum at 34 weeks gestation.

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
