# Peer review of "N-3 PUFA and Pregnancy Preserve C-Peptide in Women with Type 1 Diabetes Mellitus"

_pharmaceutics, 2021, doi:10.3390/pharmaceutics13122082_

Round 1
Reviewer 1 Report
The article under review aims to present the role of polyunsaturated fatty acids combined with pregnancy in the regulation of insulin secretion in type 1 diabetes. It is an interesting idea to highlight the synergistic effect of the two. The article is very clear and has great potential. However, in the current form, some fragments seem as if entire paragraphs were taken out of another work, which makes the overall impression that there are many reviews already available on the topic, with no need to write another one.
For instance, in lines 93-103 - there is not a single citation there (they appear far in the line 113). The same situation is in the case of the fragment between the lines 150 and 171. And between the lines 190 and 218, 253 and 275, etc. A reference should be present after every single piece of information given and the work should not make an impression of a collection of abstracts of other review papers. Some fragments in the review, especially the ones missing citations, seem to contain very basic information and way more publications on the mentioned topic are available than were mentioned in this review. Perhaps a bit more thorough literature search would help here. Otherwise, maybe some fragments could be omitted in this review, as it seems that they are entirely quoted from other review works, such as the reference 35.
Also, I feel there is no use quoting papers older than 20 years in such works unless an exceptionally meaningful piece of information is to be given.
Table 1 also could be skipped. Naming SFAs, MUSAs and PUFAs is a skill gained during a basic chemistry/biochemistry course. A summary table worthy of a review paper in a journal IF 5 and above should be way more comprehensive. I suggest adding information at least about the sources of the mentioned fatty acids and their biological roles so that the table would actually summarize anything.
Review papers are always prepared based on literature search in relevant databases and hopefully relevant articles, thus, I think the section "Materials and methods" could be skipped.
Finally, a slight editorial note. The references are given in both square brackets and in parentheses. One type should be chosen for clarity.
Author Response
Reviewer #1
Reviewer: There is no single citation in lines 93-103 (they appear far in line 113).
We have added the citations.
Reviewer: The same situation is in the case of the fragment between lines 150 and 171.
We have added the citations.
Reviewer: And between the lines 190 and 218, 253 and 275.
We have added the citations.
Reviewer: Otherwise, maybe some fragments could be omitted in this review, as it seems that they are entirely quoted from other review works, such as reference 35.
Thank you for your advice. We deleted the fragment as suggested: The human body can convert or retro-convert individual fatty acids to other fatty acids of the same series, but not to convert fatty acids from the n-3 series to the n-6 series, or vice versa. By the enzymatic action, LA is translated into long-chain n-6 PUFAs such as dihomo-γ-linolenic acid (DGLA, C20: 3n-6) and arachidonic acid (AA, 20: 4n-6). In contrast, ALA can be translated into long-chain n -3 PUFAs of which eicosapentaenoic acid (EPA, C20: 5n-3), docosapentaenoic acid (C22: 5n-3) and docosahexaenoic acid (DHA, C22: 6n-3) are predominant [35].
Reviewer: Also, I feel there is no use quoting papers older than 20 years in such works unless an exceptionally meaningful piece of information is given.
We only included papers published from 2000 to 2021.
Reviewer: Table 1 also could be skipped.
Thank you for your advice; we deleted Table 1.
Reviewer: I think the section "Materials and methods" could be skipped.
The "Materials and methods" section is changed and included in the following format: The literature was reviewed using relevant databases (Pubmed/Medline, Scopus, Science Direct). The following inclusion criteria were considered: studies that investigated hypoglycemia (hypoglycemia in pregnant women with type 1 diabetes), n-6 PUFA and n-3 PUFA (in pregnant women; in pregnant women with type 1 diabetes), supplementation of EPA and DHA (in pregnant women with type 1 diabetes) C-peptide (in pregnant women with type 1 diabetes; in nonpregnant people), an immune system in pregnancy, vitamin D in T1DM and relevant observational, randomized control trial and review articles were selected to provide information and data for the text (published from 2000 to 2021). Papers only in English were selected.
Exclusion criteria
We excluded papers with gestational and Type 2 diabetes. Articles on immunosuppressive drugs connected with C-peptide preservation (corticosteroids, azathioprine, methotrexate, cyclophosphamide, cyclosporine) were excluded.
Reviewer 2 Report
- The current form of the manuscript cannot convince the reader that any solid evidence underlies this hypothesis:
- The Material and Methods section has a limited extent and offers only scarce details about the search methodology, making it impossible to replicate
- Large sections of the paper review data unrelated to the main topic of the manuscript. By contrast, sources that support the authors’ hypothesis are mentioned briefly and, except for ref. 49 (belonging to the same authors), without any details about the results, to help the reader build an opinion by him-/herself
- Description of data in ref. 49 is confusing, by mentioning both 59.6 pmol/L and 79.7 pmol/L as the concentrations of fasting C-peptide during the first trimester of pregnancy
- Some references the authors cite to support their hypothesis seem to offer a differing perspective on this matter. Source 32, for example, concludes that the increase of C-peptide concentrations during pregnancy is due to the “transfer of C-peptide from fetal to maternal serum and is inconsistent with pregnancy-related β-cell regeneration”; contrarily, the authors use this reference to support the contrasting hypothesis. Source 33 found “no gestational changes in plasma C-peptide concentration”, but the authors also mention it in an opposite context. The main topic of sources 52-54 is completely different from the insulin secretion during pregnancy in type 1 diabetes female patients.
- Although real and severe, the matter of hypoglycaemia during pregnancy has many other causes than the one supported by the authors. None of these other causes is analysed and balanced with their current theory, which deviates from the principle of scientific impartiality. We should all remember that coincidence does not mean causality.
- What does the “Preclinical autonomy towards beta cells with progressive decline in insulin production” (page 1, lines 33-34) mean?
- Even though the general quality of the English language is acceptable, the authors should recheck the text to remedy some situations of missed or misused articles, unnatural constructs, inversed English topics, isolated typing errors or missed subjects etc.
- The following construct is unclear and should be revised: “so the most common forms of PG series 2 in the human body are eicosanoids synthesised from fatty acids n-6 (AA) and n-3 (EPA) order they interact antagonistically.” (page 6, lines 220-222)
Author Response
Reviewer 2.#
Reviewer: The Material and Methods section has a limited extent and offers only scarce details about the search methodology, making it impossible to replicate.
The section "Materials and methods" changed.
Reviewer: Large sections of the paper review data unrelated to the main topic of the manuscript. By contrast, sources that support the authors' hypothesis are mentioned briefly and, except for ref. 49 (belonging to the same authors), without any details about the results, to help the reader build an opinion by him-/herself.
Thank you for your comment. The section n-3 PUFA in C-peptide preservation in pregnant women with T1DM was rewritten, and section n-3 PUFA in C-peptide preservation in a nonpregnant patient with T1DM was added. Please note that the related references are also added, and the previous ref. 49 is now 37.
Reviewer: Description of data in ref. 49 is confusing by mentioning both 59.6 pmol/L and 79.7 pmol/L as the concentrations of fasting C-peptide during the first trimester of pregnancy.
The results can be accessed in the original article now under ref 37. We decided to rearrange the section n-3 PUFA in C-peptide preservation in pregnant women with T1DM, and it was rewritten.
Reviewer: Some references the authors cite to support their hypothesis offer a differing perspective on this matter. Source 32, for example, concludes that the increase of C-peptide concentrations during pregnancy is due to the "transfer of C-peptide from fetal to maternal serum and is inconsistent with pregnancy-related β-cell regeneration"; contrarily, the authors use this reference to support the contrasting hypothesis.
Explanation. The authors (Meek, CL.et al. previous reference 32, now reference 31) have measured maternal serum C-peptide concentrations at 12, 24, and 34 weeks of gestation in 127 pregnant women with type 1 diabetes and cord blood C-peptide concentrations. In 74 (58%) pregnant women, C-peptide was not detected; in 22 (17%), it was confirmed at the beginning of pregnancy, and in 31 (24%), it was seen in the 34th week of pregnancy. Neonates born to the mothers in whom C-peptide was detected at 34 weeks of gestation had elevated cord blood C-peptide and more frequent neonatal complications. It is worth noting, in the mothers who had higher serum C-peptide values, lower concentrations of C-peptide and glucose and lower IR HOMA-2 were found in the umbilical vein compared to mothers with lower C-peptide values.
Therefore, the higher concentration of C-peptide in the cord blood in Pattern 3 found by Meek CL et al. is evidence of poorly regulated maternal glycemia resulting in delivery of disproportionately LGA newborn and does not represent evidence C-peptide transport from fetus to mother and cause-effect of fetal macrosomia. In RCT (Horvaticek M, et al.) study did not find an association between maternal venous serum C-peptide concentration and umbilical vein, which does not mean that there is no possibility of passing a minuscule amount of C-peptide from fetus to mother.
According to other authors' research, it is plausible that pregnancy yields immunological tolerance and stimulates endogenous insulin production in women with type 1 diabetes mellitus.
It can be seen in Table 1 (Meek, CL.et al.) the authors found a higher concentration of C-peptide in the mother with a shorter duration of diabetes. Higher C-peptide concentrations were associated with lower insulin doses, as shown by the authors in Table 1. (Meek, CL.et al.). Pregnant women with higher endogenous insulin have better-regulated glycemia, fewer hypoglycemia events, reduced total insulin dose, lower incidence of macrosomia, and lower umbilical vein C-peptide, consistent with findings from Meek CL et al. Meeck's results do not contradict the results of other authors that we presented in the review article.
Reviewer: Source 33 found "no gestational changes in plasma C-peptide concentration," but the authors also mention it in an opposite context.
Explanation. Murphy, HR. et al. 2012 showed ten mothers aged 31.1 (28.7–31.7) years, with diabetes duration of 19 (13.5–24) years. The authors included a statistical analysis of C-peptide concentration with 0 pmol / L (Table 1), which explains their results. The authors appear to have had C-peptide detected in only four mothers.
Due to the small number of participants, the authors did not prove an increase in C-peptide concentration during pregnancy, which does not conflict with other authors' findings. (Previous reference 33 is now reference 32)
Reviewer: The main topic of sources 52-54 is completely different from the insulin secretion during pregnancy in type 1 diabetes female patients.
We agree. We have excluded the listed references 52-54: Madsbad, S.; Krarup, T.; Reguer, L.; Faber, O.K.; Binder, C. Effect of strict blood glucose control on residual b-cell Function in insulin-dependent diabetics. Diabetologia. 1981, 20, 530-534. Wellens, M.J.; Vollenbrock, Dekker P.; Boesten, L.S.M.; Geelhoed-Duijvestijn, P.H.; Martine, M.C.; de Vries-Velraeds, M.M.C.; Nefs, G.; Wolffenbuttel, B.H.R.; Aanstoot, H-J.; van Dijk, P.R. Residual C-peptide secretion and hypoglycemia awareness in people with type 1 diabetes. BMJ Open Diabetes Res Care. 2021, 9, e002288.
Reviewer: Although real and severe, the matter of hypoglycaemia during pregnancy has many other causes than the one supported by the authors. None of these other causes is analysed and balanced with their current theory, which deviates from the principle of scientific impartiality. We should all remember that coincidence does not mean causality.
Thank you very much for this valuable comment; we did not do it on purpose. The section "C-peptide, insulin doses, and glycemic control" has been modified.
Reviewer: What does the "Preclinical autonomy towards beta cells with progressive decline in insulin production" (page 1, lines 33-34) mean?
Corrected: autoimmunity
Reviewer 3 Report
The authors review the cross talk of n-3 polyunsaturated fatty acids, pregnancy and T1DM. The paper is interesting but at this point it requires a revision. See suggestions for revision below:PubMed is not a database, the database is MEDLINE. Correct to PubMed/MEDLINE or replace with MEDLINE.
Can you please add inclusion and exclusion criteria for the selected/excluded papers?
Add references for lines 93-103.
Lines 118-123 - please rephrase, it is difficult to understand this paragraph.
Lines 125-129 - please rephrase, it is difficult to understand this paragraph.
Lines 133 - Th1 and Th2 cells/lymphocytes
Lines 146-148 - please be more specific.
line 191 - legumes (legumes) - correct
The paper ends in an abrupt fashion. Although T1DM in pregnancy and n-3 PUFA were nicely presented, the link between T1DM and n-3 PUFA is insufficiently discussed from my point of view.
Do you recommend n-3 PUFA supplementation in pregnancy or prior to it in females with T1DM?
Please summarize the data regarding T1DM, pregnancy and n-3 PUFA in a table and design an original figure.
Pregnancy can also be associated with other micro/macronutrient deficiencies, e.g., vitamin B12, vitamin D deficiency/insufficiency. These disorders can impact the outcome of the pregnancy. In some instances, novel ultrasound parameters, such as the fetal liver, can be used to monitor the pregnancy. See the following interesting papers.
https://pubmed.ncbi.nlm.nih.gov/33001182/
https://pubmed.ncbi.nlm.nih.gov/33356450/
https://pubmed.ncbi.nlm.nih.gov/34360631/
Author Response
Reviewer 3#
Reviewer: PubMed is not a database; the database is MEDLINE. Correct to PubMed/MEDLINE or replace with MEDLINE.
Corrected.
Reviewer: Can you please add inclusion and exclusion criteria for the selected/excluded papers?
Added.
Reviewer: Add references for lines 93-103.
Added.
Reviewer: Lines 118-123 - please rephrase, it is difficult to understand this paragraph
Reviewer: Lines 125-129 - please rephrase, it is difficult to understand this paragraph.
Reviewer: Lines 133 - Th1 and Th2 cells/lymphocytes
Reviewer: Lines 146-148 - please be more specific.
Reviewer: Lines 118-123 - please rephrase, it is difficult to understand this paragraph
Reviewer: Lines 125-129 - please rephrase, it is difficult to understand this paragraph.
We accepted all comments and suggestions. The manuscript was sent for English revision to the editing service.
Reviewer: Lines 133 - Th1 and Th2 cells/lymphocytes
Reviewer: Lines 146-148 - please be more specific
Corrected:
The authors [30] have shown that C-peptide concentrations increase gradually during pregnancy in women with type 1 diabetes.
Meeck CL et al. [31] have measured maternal serum C-peptide concentrations at 12, 24, and 34 weeks of gestation in 127 pregnant women with type 1 diabetes and cord blood C-peptide concentrations. In 74 (58%) pregnant women, C-peptide was not detected; in 22 (17%), it was confirmed at the beginning of pregnancy, and in 31 (24%), it was detected in the 34th week of pregnancy. Neonates born to the mothers in whom C-peptide was detected at 34 weeks of gestation had elevated cord blood C-peptide and more frequent neonatal complications than others. Based on the results, the authors suggest a transfer of C-peptide from fetal to maternal serum without the regeneration of pregnancy-related beta cells. Pregnant women with detected C-peptide had better-regulated glycemia, fewer hypoglycemia events according to CGM, reduced total insulin dose, and lower incidence of macrosomia [31]. According to other authors' research, it is plausible that pregnancy yields immunological tolerance and stimulates endogenous insulin production in women with type 1 diabetes mellitus [30]. Due to the small number of participants, the authors [32] did not prove an increase in the concentration of C-peptide during pregnancy, which does not opposite the findings of other authors.
Reviewer: line 191 - legumes (legumes) – corrected
Reviewer: The paper ends in an abrupt fashion. Although T1DM in pregnancy and n-3 PUFA were nicely presented, the link between T1DM and n-3 PUFA is insufficiently discussed from my point of view.
The section was rewritten.
Reviewer: Do you recommend n-3 PUFA supplementation in pregnancy or before it in females with T1DM?
Yes, we would like to recommend n-3 PUFA in preconception and during pregnancy and lactation.
Reviewer: Please summarize the data regarding T1DM, pregnancy, and n-3 PUFA in a table and design an original figure.
The data regarding T1DM pregnancy C-peptide and n-3 PUFA are summarized in Table 1.
Reviewer: Pregnancy can also be associated with other micro/macronutrient deficiencies, e.g., vitamin B12, vitamin D deficiency/insufficiency. These disorders can impact the outcome of the pregnancy.
We added the impact of vitamin D on T1DM.

Reviewer 4 Report
The manuscript entitled: “The combination of n-3 polyunsaturated fatty acids and pregnancy stimulates endogenous insulin production in women with type 1 diabetes” is a review dealing an intersting topic, nonetheless the text is poorly written and should need a better assessment to contribute to the area of interest. The end points should be more clearly assessed and data presented in a more clear form with reference to the manuscript title.The title is too long and should better focus the end points of the proposed review. The Figure 1 should need a comment on the axis to better assess the context. The Materials and Methods section needs to be improved and to explain better the criteria used in selecting the literature data to be commented in the review (e.g. the year range, etc.). How the Authors selected the papers and do discriminate the ones to be excluded from the data analysis. The role of fatty acid sas per the manuscript title should be evidenced based on clinical data, e.g. In form a Table grouping the clinical studies, if any, relevant to the topic. The Table 1 is not relevant to the context: why the Authors consider it necessary to assess the context of the manuscript? Paragraph 4.1 seems too long and should be also explained why it is necessary in the context of the proposed review paper. The topic of the review starts to be considered only at page 6: please explain. The References too dated should be avoided (see the one dated 1989, etc.) in favor of more recent literature data. In conclusion, the manuscript in its present form cannot be recommended for publication in the Journal.
Author Response
Reviewer 4#
Recommendation
Reviewer: nonetheless, the text is poorly written and should need a better assessment to contribute to the area of interest.
Thanks to the reviewers' advice, the authors corrected the mistakes and expanded the text.
Reviewer: The endpoints should be more clearly assessed and data presented in a more explicit form concerning the manuscript title.
The manuscript has been significantly modified.
Reviewer: The title is too long and should better focus on the endpoints of the proposed review.
The new title is: N-3 PUFA and pregnancy preserve C-peptide in women with type 1 diabetes mellitus
Reviewer: Figure 1 should need a comment on the axis to assess the context better.
It has been done.
Reviewer: The Materials and Methods section needs to be improved and explain the criteria used in selecting the literature data to be commented on in the review (e.g., the year range, etc.). How the chosen authors the papers and do discriminate the ones to be excluded from the data analysis.
It has been done.
Reviewer: The role of fatty acid as per the manuscript title should be evidenced-based on clinical data, e.g., In the form of a Table grouping the clinical studies, if any, relevant to the topic.
The data regarding T1DM pregnancy, C-peptide, and n-3 PUFA are summarized in Table 1 ( please be aware that this is new Table 1 ).
Reviewer: Table 1 is not relevant to the context: why do the Authors consider it necessary to assess the context of the manuscript?
Table 1 was excluded.
Reviewer: Paragraph 4.1 seems too long and should also explain why it is necessary for the proposed review paper.
We have shortened paragraph 4.1
Reviewer: The topic of the review starts to be considered only on page 6: please explain.
We have rearranged the manuscript and hope to tackle the topic successfully.
Reviewer: The References too dated should be avoided (see the one dated 1989, etc.) in favor of more recent literature data.
All references written before the year 2000 have been removed from the manuscript. The references published from 2000 to 2021 were selected.

Round 2
Reviewer 1 Report
The manuscript has been sufficiently corrected and I have no further comments.
Reviewer 2 Report
The authors have overall chosen to maintain their one-sided approach on the matter, even though proof supporting the recovery of β-cell endogenous insulin secretion under the influence of various favourable factors is, at this moment, merely circumstantial and not categorical. An impartial approach, equally analysing all hypotheses and avoiding biases, would have been advisable and would have provided a superior value to their endeavour. Our own beliefs should not drive the way of presenting scientific pro and con evidence.
For instance, glucose control is often stricter during pregnancy to avoid foetal and maternal disruption. Or, the simple improvement of plasma glucose values reduces the glucotoxicity factor, thus releasing the pressure from the still surviving β cells and allowing them to secrete a supplemental amount of insulin. Regrettably, the manuscript does not analyse this perspective. The benefits of vitamin D, which is at present another “fashionable” referral, but is supported by an uneven body of evidence, are also described in an inequitable approach.
Reviewer 3 Report
The authors have addressed my comments.
Reviewer 4 Report
The manuscript has been modified and improved. The comments have been addressed. There are still minor drawbacks to assess, e.g. the shortenings should be defined in full at their first use in the manuscript and the English should be revised for better readability. The end points of the manuscript should be better set at the very beginning and expolited better as for example in the Concusion section which should include perspective point of view of the Authors regarding the impact of the proposed paper in the area of interest.